# Translation and Validation of the Indicators of Quality Nursing Work Environments in the Portuguese Cultural Context

**DOI:** 10.3390/ijerph191912313

**Published:** 2022-09-28

**Authors:** Eliana Sousa, Chiou-Fen Lin, Filomena Gaspar, Pedro Lucas

**Affiliations:** 1Nursing Research, Innovation and Development Centre of Lisbon (CIDNUR), Nursing School of Lisbon, 1600-190 Lisbon, Portugal; 2Tâmega e Sousa Hospital Center, Avenida do Hospital Padre Américo 210, Guilhufe, 4560-136 Penafiel, Portugal; 3School of Nursing, National Taipei University of Nursing and Health Sciences, Taipei City 112, Taiwan; 4Department of Nursing, Shuang Ho Hospital, Taipei Medical University, New Taipei City 235, Taiwan

**Keywords:** indicators, management, nursing, psychometric properties, quality of care, work environment, validation study

## Abstract

The Indicators of Quality Nursing Work Environment scale (IQN-WE) with 65 items, developed by Chiou-Fen Lin, Meei-Shiow Lu, and Hsiu-Ying Huang in 2016, aimed to create a set of quality indicators of the nursing work environment. The translation and validation for the Portuguese cultural context of the IQN-WE scale was performed in this study. Objectives: culturally and linguistically adapt the IQN-WE scale, originating from the Portuguese version, and evaluate its psychometric characteristics. Methods: it is a descriptive, cross-sectional, observational and quantitative study. The IQN-WE scale was validated in a sample of 542 nurses belonging to a hospital center in Portugal. The study obtained a response of 21.69%, of whom 78.0% were women and 22% men. The mean age of the sample was 39 years and a standard deviation of 8.1 years. All nurses who work have a bachelor’s degree, and 13.5% have a master’s degree. Results: the study obtained an instrument with a total explained variance of 52.67% and KMO = 0.843. It found a strong-to-moderate linear correlation matrix between the dimensions. The pre-test and the team of experts ensured the content validity. The determination of internal consistency guaranteed reliability, with 0.95. Confirmatory factor analysis validated the construct. The factorial model presented a goodness of fit index, with five factors. Conclusion: the study achieved an instrument with 50 items in five dimensions: Team Support and Professional Development; Team Organization and Management; Safe Nursing Work Environment; Information Systems and Risk Control; Salary and Welfare. The IQN-WE-PT scale proved to be an appropriate instrument to be applied in health organizations in Portugal.

## 1. Introduction

The investment in improving the nursing work environment has an objective of improving the working conditions of nurses, which impacts the quality of nursing care provided. For this, it is necessary to increase the number and quality of nurses and improve the management of organizations, organizational culture, and professional training [1,2,3]. The quality of nursing care is enhanced when nurses experience a good quality of life at work and a safe nursing work environment because these aspects are fundamental in the provision of care in health organizations [1].

The nursing work environment is critical to the success of health systems [2] and is associated with the quality of care, patient safety, professional satisfaction, efficiency of organizations and effectiveness of care for patients [3,4,5,6]. The nursing work environment is established by the organizational characteristics of a work situation that constrain or facilitate the professional practice of nurses [4].

A positive nursing work environment can lead the effectiveness of health organizations to improved patient outcomes, being an essential condition for increasing nurses’ satisfaction [2,4,7], which is crucial to keep the teams with safe allocations and reduce the turnover of nurses [5,6]. A favorable nursing work environment has positive characteristics, such as the appropriate of human and material resources, the quality of care, the participation of nurses in the governance of organizations, good relations between different professional groups and provision of nursing care [2,4,5,8].

On the other hand, unfavorable work environments, where there are levels of stress and high workload, tend to provide employees with disorders, such as fatigue, turnover, absenteeism, lower concentration, anxiety and burnout, among others [5,9].

Nurse managers play a key role in creating a supportive nursing work environment [10] and promoting quality care [3,5,8]. They can also facilitate the professional development of nurses and future managers [11]. Nursing leadership plays a central office in quality patient care, which requires four essential activities: strengthening intra- and interprofessional relationships, facilitating effective continuous communication, building and maintaining teams, and peer involvement [3,8]. Nurses as leaders are essential to improve communication with and among the team to achieve goals, aiming at quality of care, patient safety and innovation in health [3,12].

The quality of nursing care is part of the provision of safe nursing care, based on nursing standards, such as patient satisfaction and safety [13]. The quality of care is an indispensable component in the profession and can affect to the relationship between the nurse and the patient, which depends on various aspects, particularly the nursing work environment [5,14]. The nursing work environment is related to and influences the quality of care, with the objective of all interventions to obtain results for the patient and the professional.

Nurse managers have a fundamental role in increasing the autonomy and involvement of their team to promote organizational responsibility and commitment, as well as increasing the motivational quality of work characteristics, creating a favorable nursing work environment and boosting the quality of nursing care [15,16,17,18], enabling the necessary conditions for the professional development of nurses and future managers [19].

The Indicators of Quality Nursing Work Environment scale (IQN-WE) developed by Chiou-Fen Lin, Meei-Shiow Lu and Hsiu-Ying Huang in 2016, contains a safe work environment, team quality, team training, professional development, and nursing care support. The IQN-WE scale promotes the relevance of professional specialization and team collaboration. In addition, the IQN-WE scale adds relevant factors, such as salary and welfare, simplification of tasks and nursing information systems [1].

The IQN-WE scale consists of eight subscales with 65 items, each item can be classified using the 4-point Likert scale (1 point—very unsuitable to 4 points—very suitable). The total variation of results can range from 65 to 260 points and the lower the score, the more unfavorable the quality indicators of the nursing work environment will be. This scale aimed to develop a set of quality indicators of the nursing work environment in Taiwan [1].

This study aimed to translate and adapt to the Portuguese cultural context and psychometrically validate the scale IQN-WE.

## 2. Materials and Methods

### 2.1. Study Design

This study is cross-sectional, observational and quantitative.

### 2.2. Participants and Procedure

The research obtained the data between May 2021 and June 2021. The author of the original IQN-WE scale authorized the study. Nurses who worked in a hospital center in Portugal received the questionnaire by email. The data collection instrument was composed of an introduction with the objectives of the study, the informed consent form for participation, a sociodemographic and professional characterization (gender, age, academic qualifications, years of complete professional practice, years of professional practice in the organization/institution, professional category, unit where the person worked, and performance appraisals) and the IQN-WE scale.

The original IQN-WE consists of 65 items. These items are rated on a Likert-type scale with four possible responses from 1 to 4 (1 = very suitable to 4 = very unsuitable).

The study’s inclusion criteria were nurses who worked in a hospital in Portugal and gave permission to participate in the questionnaire. Researchers submitted 750 questionnaires and received responses from 542 nurses.

### 2.3. Analysis

The research performed a comparative and descriptive analysis of the variables. They were described by mode, standard deviation and mean.

The exploratory factor analysis (EFA) technique was used to confirm structural patterns [20]. Researchers extract the principal components using the Varimax rotation method to perform EFA. The study, in the analysis of the adequacy of the factors, used the Kaiser–Meyer–Olkin (KMO) test, whose value should be greater than 0.5 and the Bartlett sphericity test, which indicates the adequacy of the data for factor analysis. It analyzed the total variance explained by the results.

Through the reliability of the instrument and internal consistency, the study determined the Cronbach’s Alpha coefficient (α), which can range from 1 to 0, where 0.70 is the minimum number for acceptable reliability [21].

The evaluation of the quality of the factor model fit was performed by analyzing confirmatory factor analysis (CFA). The maximum likelihood method performed the CFA, which assumes multivariate normality and the absence of outliers. The asymmetry coefficient (Sk) and kurtosis (Ku) analyzed the normal distribution of the variables. The study evaluated the outliers using the Mahalanobis squared distance (D^2^) [20].

The study used the following indices to evaluate the overall adequacy of the model in the CFA: the comparative goodness of fit index (CFI), where a number greater than 0.90 indicates a goodness of fit, the adequacy index (GFI) and root mean square error of approximation (RMSEA) index with an acceptable score between 0.08 and 0.05 [20].

The best fit is represented in the model with the lowest expected cross-validation index (ECVI). The study handled the modification indices produced in AMOS, as well as theoretical considerations [20].

The study tested CFA using convergent and factorial validity. After confirmation of the multivariate normality, it experimented the factorial validity with maximum likelihood estimation.

The factor model presented is considered valid when all items have a factor load higher than 0.4 [21]. Statistical software SPSS Statistics version 26.0 and AMOS (IBM Corp, Armonk, NY, USA) performed all analyses.

### 2.4. Ethical Considerations

For the cultural, linguistic adaptation and psychometric validation of the IQN-WE scale, the study obtained the necessary ethical and legal authorizations to carry out this research: formal approval from the author of the original scale and authorization from the Ethics Committee and the Board of Directors of the hospital in question (PI nº09/15-04-2021). After the request was accepted, we contacted the nurse director of the hospital center to disseminate the scale via institutional email to nurses working in all services of the hospital center. This study guarantees compliance with the confidentiality and anonymity rights of the participants, and the protection of their data following the General Regulation on Data Protection 2016/679 of the European Union.

### 2.5. Translation Process and Cultural Adaptation of the IQN-WE Scale

The IQN-WE-PT scale followed the methodological guidelines regarding the development of the linguistic and conceptual equivalence [22,23].

In the first phase of the translation process, two independent translators—one bilingual, fluent in English and Portuguese, with nursing literacy, and another certified translator without health literacy, who issued a translation certificate—translated the original questionnaire from English into Portuguese and verified its semantic equivalence. The two translators respected the idiomatic, semantic and experiential equivalence of the questionnaire translation, maintaining the conceptual meaning of the original version.

Then, in a second step, a group of experts carried out an analysis of the two versions, evaluating each item individually with the aim of originating a consensus version. Two items were eliminated with the authorization of the original author at this stage because they did not fit into the Portuguese context. In a third step, the retroversion was carried out by a third translator, fluent in English and Portuguese, unaware of the original version, creating the retroverted version. In the next step, the study compared the items of the original scale with those of the retroverted scale, item by item, to investigate whether there were doubts or disparities and verify semantic, idiomatic and conceptual equivalence.

After the adaptations, the translated version of IQN-WE was obtained. The study performed the pre-test to verify the equivalence/validity of the content. It applied the pre-test to a sample of 27 nurses, in April 2021, with a filling time that varied between 10 and 15 min. All answered that the questionnaire did not give rise to doubts, originating the final version of the IQN-WE-PT scale.

## 3. Results

The investigation obtained a response of 21.69%, and the sample consisted of 542 nurses in total, of whom 78.0% were women. The mean age of the sample was 39 years. All nurses who work have a bachelor’s degree, and 13.5% have a master’s degree. Nurses with a post-graduate academic degree of specialization represented 28.4% of the sample, and 15.6% of nurses have post-graduate degrees. The average total professional experience was 16 years, and the average professional experience in the organization was 13 years.

### 3.1. Exploratory Factor Analysis

The Bartlett sphericity test had 0.000 and the KMO index 0.822, both good values for principal component analysis [21].

The IQN-WE-PT identified five components that explain 52.67% of the total variance. Items 1, 2, 3, 12, 13, 17, 18, 19, 24, 28, 49, 58 and 63 did not meet the factor load criteria and were excluded for this reason. The final scale covered of 50 items in five components: “team support and professional development,” with 16 items; “team organization and management,” with 13 items; “safe nursing work environment,” with 7 items; “Information Systems and Risk Control,” with 7 items; “salary and welfare,” with 7 items (Table 1).

### 3.2. Reliability Analysis

The reliability analysis for the full scale had a Cronbach’s alpha of 0.953, which is adequate and corresponds to high internal consistency and excellent reliability [21]. Table 1 presents Cronbach’s alpha values in each of the component were higher than 0.80.

### 3.3. Confirmatory Factor Analysis

Confirmatory factor analysis (CFA) was completed using the five-factor structure of the IQN-WE-PT scale. The CFA model showed a moderate goodness of fit index (2,545,165/1164 = 2.817; CFI = 0.669; GFI = 0.599; RMSEA = 0.092; P [rmsea ≤ 0.05] < 0.001; MECVI = 20,674).

The study evaluated values of Mahalanobis distances that suggested that they were multivariate outliers (p1 and p2 < 0.001) to improve the global model fit, being excluded from the CFA.

Figure 1 presents the values of the model in terms of local fit, including the standardized factor weights and the individual reliability of each item. All items have standardized factor weights (λ) higher than 0.5 and individual reliabilities (λ 2) higher than 0.25, except for e7, e10, e13, e36, e40, e41, and e42, with values between 0.10 and 0.24, thus revealing that the factors present factorial validity in all their factors: “Team Support and Professional Development;” “Team Organization and Management;” “Safe Nursing Work Environment;” “Information Systems and Risk Control;” “Salary and Welfare.”

As an indicator of convergent validity, the AVE proved to be acceptable for the SNWE and low for the remaining factors. The Penta factorial model presented a goodness of fit index (2,033,593/1119 = 1.817; CFI = 0.781; GFI = 0.675; RMSEA = 0.076; P [rmsea ≤ 0.05] < 0.001; MECVI = 18,031).

## 4. Discussion

The IQN-WE scale was translated into Portuguese, and the instrument proved to be reliable and valid for evaluating the quality indicators of the nursing work environment.

Content validity was ensured throughout the instrument’s construction, including a pre-test and a team of experts. Translation in the area under study requires that the language used is translated carefully, so that certain concepts are equivalent among the various countries that use the same original scale. The stages of translation, retroversion, and analysis were essential to produce and maintain the objectivity of the process of translation and cultural adaptation of the IQN-WE. The IQN-WE-PT scale has undergone some changes concerning the original, most likely because it is a different cultural reality, with distinct health care use and health policies [24] due to the existence of a National Health Service. Confirmatory factor analysis determined the validity of the construct. The IQN-WE-PT scale demonstrated excellent internal consistency (Cronbach’s α = 0.95), with 50 items and five dimensions.

Two items (2.1; 2.2) were excluded from the original scale at the translation stage because they did not fit into the Portuguese cultural context. In the factor analysis, 13 items were eliminated (1, 2, 3, 12, 13, 17, 18, 19, 24, 28, 49, 58 and 63) in relation to the original questionnaire.

The sample in this study shares similar sociodemographic characteristics to the participants of the original study [1]; the study by Vranada (2016) [25]; the study developed by the same author of the original IQN-WE scale and his team, published in 2020 [26]; and the study by Richards (2021) [27], which used the same scale. The age range of the sample is higher than most studies, which is between 31 and 40 years. The majority of the sample presents a comprehensive professional experience between months and 15 years, which converges with the studies of Vranada and the two studies of Lin and his team [1,25,26]. Only the original study [1] presented data on professional experience in the organization, which agrees with this study, where the sample has an average value of 13 years. Regarding the professional category, the sample agrees with international studies, in which the majority has the professional category of nurses. The female gender presents better adequacy of the quality indicators of the nursing work environment than the male gender. Concerning complete professional experience, nurses with more professional experience tend to have lower values of the adequacy of the quality indicators of the nursing work environment. In the professional category, nurse managers had higher scores than the rest of the nurses, which indicates that nurse managers have a different perspective regarding the adequacy of the indicators.

After carrying out the cultural adaptation process and the psychometric analysis of the IQN-WE-PT scale for the Portuguese population, the results obtained indicate that this questionnaire can be used in health organizations in Portugal where nurses provide care.

### Limitations

The questionnaire application was carried out only in a hospital center, since the largest number of nurses work in a hospital context in Portugal, though it would bring more benefits if it were implemented at all over country and in different areas of care practice. Another limitation of this study was the number of responses obtained to the questionnaires delivered, which may have fallen short of the expectations to all the professional and organizational constraints that nurses experienced during the COVID-19 pandemic, despite complying with the number of responses to a validation study [20].

## 5. Conclusions

The present study showed an adequate structure of five factors of the IQN-WE-PT scale in a sample formed by hospital nurses. The study obtained results that indicate that the IQN-WE-PT scale can be used safely, since it fulfilled all the methodological steps systematically and presents good results regarding its validity and reliability.

The study proves that the IQN-WE-PT scale contributes to new quality indicators of the nursing work environment to improve patient safety, professional satisfaction, efficiency of organizations and effectiveness of care for patients. This instrument makes it possible to provide an essential tool for nurse managers that, when applied to clinical practice nurses, makes it possible to obtain results that indicate nurses’ perception of the quality of nursing care and the nursing work environment, that is, which positive points or improvements to the nursing work environment in each service or organization are required in order to have more health gains.

The IQN-WE-PT scale has good psychometric characteristics for the Portuguese population, so the study suggests that this scale be applied in future studies at all levels of care in health systems.

The results obtained at the end of the study showed that the IQN-WE-PT scale can be used in nursing management, nursing practice, and nursing research. This study provides information that indicates nurses’ perception of the quality care and the nursing work environment of each health unit or organization, which points to improvement in this area and offers contributions to the academic community by providing a validated data collection instrument for the Portuguese health organizations.

Finally, this instrument offers a vital contribution to the profession and health policies, since it collaborates to the improve the quality of care and the environment of nursing practice in the various work contexts of nurses.

## Figures and Tables

**Figure 1 ijerph-19-12313-f001:**
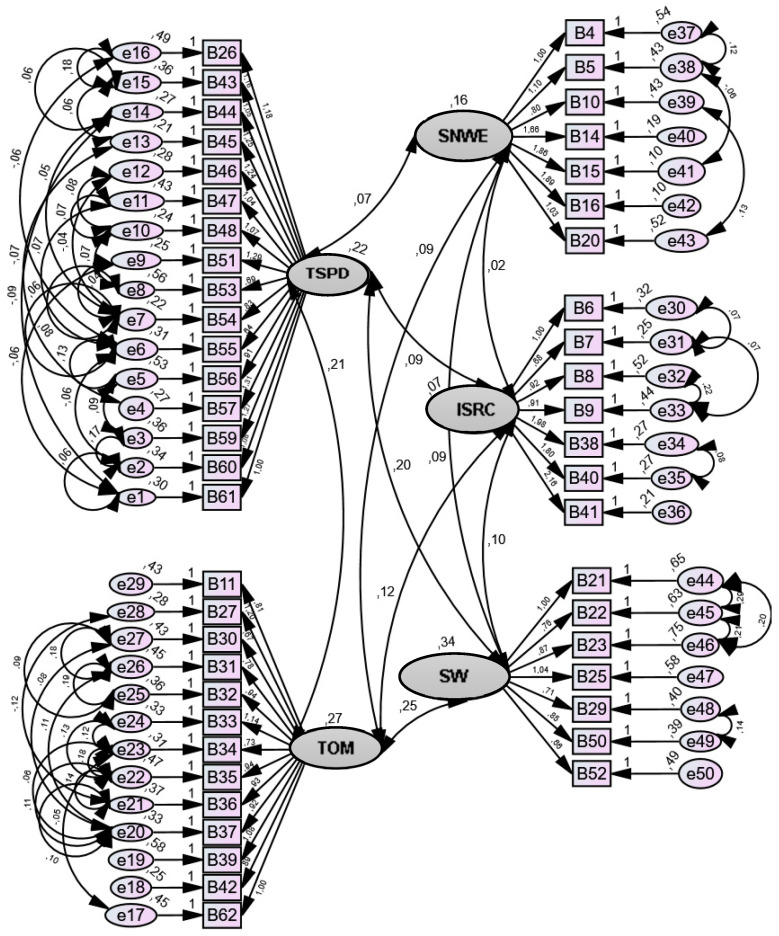
Five-factor model of the IQN-WE-PT scale.

**Table 1 ijerph-19-12313-t001:** Components of the IQN-WE-PT scale.

Item	Component				
	Team Support and Professional Development	Team Organization and Management	Safe Nursing Work Environment	Information Systems and Risk Control	Salary and Welfare
26	0.43				
43	0.62				
44	0.62				
45	0.65				
46	0.76				
47	0.65				
48	0.75				
51	0.65				
53	0.57				
54	0.56				
55	0.59				
56	0.53				
57	0.62				
59	0.65				
60	0.55				
61	0.54				
11		0.42			
27		0.46			
30		0.45			
31		0.57			
32		0.61			
33		0.52			
34		0.51			
35		0.63			
36		0.54			
37		0.68			
39		0.65			
42		0.49			
62		0.51			
4			0.43		
5			0.53		
10			0.57		
14			0.81		
15			0.82		
16			0.83		
20			0.47		
6				0.61	
7				0.62	
8				0.55	
9				0.55	
38				0.46	
40				0.53	
41				0.50	
21					0.58
22					0.55
23					0.58
25					0.40
29					0.55
50					0.63
52					0.49
AVE	0.38	0.31	0.44	0.31	0.30
Alpha of Cronbach	0.92	0.90	0.85	0.82	0.80

## Data Availability

Restrictions apply to the availability of these data. Data were obtained from a third party and are available with the permission of the third party.

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
