# Peer review of "Translation and Validation of the Indicators of Quality Nursing Work Environments in the Portuguese Cultural Context"

_ijerph, 2022, doi:10.3390/ijerph191912313_

Round 1
Reviewer 1 Report
Thank you for the opportunity to review this study entitled “Translation and Validation of the Indicators of Quality Nursing Work Environments in the Portuguese cultural context.” (ijerph-1933619).
The research aimed at translating and validating the Indicators of Quality Nursing Work Environments (IQN-WE) in the Portuguese cultural context. The study involved a sample of 4 542 nurses.
In my opinion, the research topic is relevant, and the study is interesting. Parallelly, some issues need to be addressed before the paper will be suitable for publication.
1. In the abstract, the information about the sample should be deepened (N? Mean age and SD? Percentage of men and women?) for both the two groups (flight and train attendant students; general population) to provide a clear picture of what will be presented in the paper.
2. Abstract: Please remove the numerical indices to improve the readability.
3. The IQN-WE should be described more in-depth
4. Please provide hypotheses.
5. The “2.2. Method” section should change the name to “Participants and procedure”.
6. The absence of an assessment of convergent and divergent validity is a strong limitation of this research that should be highlighted.
7. The practical implications of these findings should be described in more depth in the “Conclusion” section.
Best wishes
Reviewer 2 Report
Dear authors!
In my opinion, this manuscript is interesting. Thank you for the opportunity of reviewing it. However, there are some points to be considered before publication.
Introduction:
Please, give more than one citation in first two paragraphs (line 33-42).
Also, too much space was taken up by explaining the nursing work environment and the role of the nurse manager, and less by explaining the importance of the scale and explaining the scale itself. Would you please improve that - explain the scale better.
METHODS:
Please provide additional information on how to recruit participants and more demographic details of your participants. Please consider adding such information as a description of any exclusion criteria that were applied to participant recruitment, a description of how participants were recruited, and descriptions of where participants were recruited and where the research took place.
RESULTS:
The results section is presented clearly and follows accepted standards followed by excellent statistical analysis. Great that the Warimax rotation method was used to perform EFA.
DISCUSSION:
It would be good to expand it by a few paragraphs. Limitation is that research was carried out only in a hospital center, but maybe you should refer to the fact that ( I assume ) the largest number of nurses work in the hospital system in Portugal.
